# High VEGFR3 Expression Reduces Doxorubicin Efficacy in Triple-Negative Breast Cancer

**DOI:** 10.3390/ijms24043601

**Published:** 2023-02-10

**Authors:** Sandra Torres-Ruiz, Eduardo Tormo, Iris Garrido-Cano, Ana Lameirinhas, Federico Rojo, Juan Madoz-Gúrpide, Octavio Burgués, Cristina Hernando, Begoña Bermejo, María Teresa Martínez, Ana Lluch, Juan Miguel Cejalvo, Pilar Eroles

**Affiliations:** 1Biomedical Research Institute INCLIVA, 46010 Valencia, Spain; 2Center for Biomedical Network Research on Cancer (CIBERONC), 28029 Madrid, Spain; 3Department of Pathology, Fundación Jiménez Díaz, 28040 Madrid, Spain; 4Department of Pathology, Hospital Clínico Universitario de Valencia, 46010 Valencia, Spain; 5Department of Medical Oncology, Hospital Clínico Universitario de Valencia, 46010 Valencia, Spain; 6Department of Medicine, Universidad de Valencia, 46010 Valencia, Spain; 7Department of Physiology, Universidad de Valencia, 46010 Valencia, Spain; 8Department of Biotechnology, Universidad Politécnica de Valencia, 46022 Valencia, Spain

**Keywords:** VEGFR3, triple-negative breast cancer, doxorubicin

## Abstract

Due to the lack of specific targets, cytotoxic chemotherapy still represents the common standard treatment for triple-negative breast patients. Despite the harmful effect of chemotherapy on tumor cells, there is evidence that treatment could modulate the tumor microenvironment in a way favoring the propagation of the tumor. In addition, the lymphangiogenesis process and its factors could be involved in this counter-therapeutic event. In our study, we have evaluated the expression of the main lymphangiogenic receptor VEGFR3 in two triple-negative breast cancer in vitro models, resistant or not to doxorubicin treatment. The expression of the receptor, at mRNA and protein levels, was higher in doxorubicin-resistant cells than in parental cells. In addition, we confirmed the upregulation of VEGFR3 levels after a short treatment with doxorubicin. Furthermore, VEGFR3 silencing reduced cell proliferation and migration capacities in both cell lines. Interestingly, high VEGFR3 expression was significantly positively correlated with worse survival in patients treated with chemotherapy. Furthermore, we have found that patients with high expression of VEGFR3 present shorter relapse-free survival than patients with low levels of the receptor. In conclusion, elevated VEGFR3 levels correlate with poor survival in patients and with reduced doxorubicin treatment efficacy in vitro. Our results suggest that the levels of this receptor could be a potential marker of meager doxorubicin response. Consequently, our results suggest that the combination of chemotherapy and VEGFR3 blockage could be a potentially useful therapeutic strategy for the treatment of triple-negative breast cancer.

## 1. Introduction

Triple-negative breast cancer (TNBC) represents approximately 15% of all breast carcinomas. This molecular subtype is characterized by a null or minimal expression of estrogen receptor, progesterone receptor, and human epidermal growth factor receptor 2 and is usually associated with poorer prognosis and higher recurrence rates compared to other molecular subtypes of breast cancer (BC) [1,2].

Receptor-positive BC has specific clinical therapeutic regimens available; however, no specific therapies have been achieved for the treatment of TNBC [3,4], despite the fact that numerous investigations are being carried out to define new therapeutic options [5,6,7]. Thus, cytotoxic chemotherapy still represents the common standard treatment for this group of patients [4]. However, despite the efforts, there is a high number of TNBC patients for whom chemotherapy treatment is not effective, which translates into continued tumor spread and metastasis. It has been described that lymphangiogenesis and sustained angiogenesis are important steps in tumor progression and that metastasis can occur partially through tumor-associated lymphatic vessels, and the lymphatic system is an important prognostic indicator of BC progression [1,8,9]. Basically, the tumor can induce its own network of vessels, which in turn interact with the surrounding ones. This may serve as an additional mechanism by which tumor cells obtain oxygen and nutrients to survive, especially in poorly vascularized regions of the tumor. The lymphangiogenic factor vascular endothelial growth factor C (VEGFC) has been shown to be able to induce the formation of lymphatic vessels in and around tumors through activation by binding to the vascular endothelial growth factor receptor 3 (VEGFR3) in lymphatic endothelial cells, thus improving metastatic spread through the lymphatic vessels [10,11,12,13,14,15].

VEGFR3 is initially expressed in the entire embryonic endothelium, intervening in the development of the vasculature, but its expression decreases during development and is largely limited to the lymphatic endothelium in adult tissues. [12]. Nevertheless, it has been observed that VEGFR3 overexpression in different tumors is associated with increased proliferation, invasion, and chemoresistance [16,17,18,19].

Doxorubicin is a standard chemotherapy treatment used against TNBC, but its effects on lymphangiogenesis are not well described. In this study, we analyze the effect of doxorubicin on VEGFR3, which modulates lymphangiogenesis-mediated tumor growth. The counter-therapeutic effect of this commonly used chemotherapy provides a new approach to how doxorubicin, surprisingly, can influence the promotion of cancer spread, and how these activating changes can be decreased by blockade of VEGFR3 signaling.

## 2. Results

### 2.1. Doxorubicin Treatment Enhances VEGFR3 Gene Expression in MDA-MB-231 and MDA-MB-468 but Not in MCF7 Cells

In order to assess the effect of doxorubicin on cell proliferation, two TNBC cell lines (MDA-MB-231 and MDA-MB-468) and one luminal A BC cell line (MCF7) were exposed for different time periods to 5 μM doxorubicin. All cell lines showed a significant drop in proliferation in a time-dependent manner (Figure 1A): after 24 h of treatment, viability was around 50% (*p* = 0.0091 for MDA-MB-231, *p* = 0.0102 for MDA-MB-468, and *p* = 0.0014 for MCF7); after 48 h, viability dropped to 20–30% (*p* = 0.0043 for MDA-MB-231, *p* = 0.0011 for MDA-MB-468, and *p* = 0.0004 for MCF7), and at 72 h, the percentage of viable cells was about 5–15% of control (*p* = 0.0009 for MDA-MB-231, *p* = 0.0006 for MDA-MB-468, and *p* = 0.0003 for MCF7). To understand the regulation and effect of VEGFR3 in the different BC cell lines upon doxorubicin treatment, we performed a comprehensive analysis of the mRNA expression using real-time quantitative PCR (RT-qPCR). An increase in the mRNA levels of *VEGFR3* was detected in MDA-MB-231 (*p* = 0.0002) and MDA-MB-468 cells (*p* = 0.0033) after 24 h of treatment. However, changes in the MCF7 cell line were not observed between treated and non-treated cells (Figure 1B). These results suggest that doxorubicin increases VEGFR3 expression specifically in the TNBC subtype, which could have a role in the lymphangiogenic process.

### 2.2. VEGFR3 Protein Is Overexpressed after Doxorubicin Treatment in MDA-MB-231 Cell Line

The evaluation of VEGFR3 protein expression by Western blot supported mRNA results since the protein became overexpressed after doxorubicin treatment in MDA-MB-231 cells. However, we were not able to detect VEGFR3 protein in MDA-MB-468 or MCF7 cells (Figure 1C and Appendix AA). An alternative analysis of VEGFR3 protein expression through immunofluorescence microscopy was carried out. VEGFR3 positive-immunoreactivity was strongly detected in MDA-MB-231 cells and weakly observed in MDA-MB-468 cells (only in cytoplasm) under doxorubicin treatment. In contrast, no VEGFR3 staining was detected in MCF7 cells (Figure 1D). In addition, since VEGFR2 and VEGFR3 form heterodimers, an immunoprecipitation assay was performed to find out if there was an increase in both receptors. The results showed strong protein overexpression of VEGFR3 in the MDA-MB-231 cell line and weak protein overexpression of VEGFR3 in the MDA-MB-468 cell line after doxorubicin treatment; no changes were observed in the MCF7 cell line. VEGFR2 did not show overexpression after treatment in any of the cell lines (Figure 1E and Appendix AB,C).

### 2.3. VEGFR3 Modulates Response to Doxorubicin in Both Sensitive and Resistant TNBC Cell Models

Basal and post-treatment VEGFR3 mRNA levels were measured in doxorubicin-sensitive MDA-MB-231 and doxorubicin-resistant MDA-MB-231R cell lines. Results showed that doxorubicin treatment induced overexpression of the VEGFR3 gene in parental MDA-MB-231 (*p* = 0.0008), while mRNA levels were found to be constitutively overexpressed in the resistant MDA-MB-231R cells (*p* = 0.0001) and were slightly increased by doxorubicin treatment (*p* = 0.0003). In addition, VEGFR3 silencer (siVEGFR3) was able to downregulate the gene expression in both cell lines independently of the treatment (*p* = 0.0281 for MDA-MB-231 and *p* < 0.0001 for MDA-MB-231R) (Figure 2A). Those results were also confirmed through protein expression analysis (Figure 2B and Appendix AD).

Regarding functional effects, silencing of VEGFR3 through transient transfection of siRNA yielded an effect on proliferation in MDA-MB-231 and MDA-MB-231R cell lines after doxorubicin treatment. While parental cells suffered a 51.35% decrease in proliferation under standard treatment conditions (24 h, 5 uM) (*p* = 0.0005), the resistant cells showed an 11% reduction compared to the control (*p* = 0.0032). In addition, silencing of VEGFR3 decreased proliferation by itself (*p* = 0.0030 for MDA-MB-231 and *p* = 0.0052 for MDA-MB-231R), and furthermore, this drop was more evident when silencing was combined with doxorubicin treatment in both cell line models (*p* = 0.0002 for MDA-MB-231 and *p* < 0.0001 for MDA-MB-231R) (Figure 2C).

In parallel, a wound-healing assay was conducted to assess cell migration. On the one hand, the MDA-MB-231 cell line was able to achieve 40% wound closure under normal conditions and when transfected with the control siRNA (siSCR). However, doxorubicin treatment (14.67% wound closure, *p* = 0.0017 vs. control), VEGFR3 silencing (8.25% wound closure, *p* = 0.0014 vs. siSCR), or a combination of both (0.38% wound closure, *p* = 0.0017 vs. siSCR + doxorubicin) prevented the cells from closing the groove. On the other hand, the MDA-MB-231R cell line was able to achieve more than 70% wound closure under normal conditions and after treatment with doxorubicin. However, VEGFR3 silencing by itself (7.64% wound closure, *p* = 0.0005 vs. siSCR) or in combination with treatment (2.27% wound closure, *p* < 0.0001 vs. siSCR + doxorubicin) affected the migratory capacity of the cells, preventing the closure of the wound. In summary, VEGFR3 silencing produced a loss of motility in both sensitive and resistant cell lines (Figure 2D and Appendix A).

### 2.4. Overexpression of VEGFR3 as a Potential Prognosis Biomarker

VEGFR3 mRNA expression was measured in anthracycline-treated TNBC-paired samples through RT-qPCR. For the analysis, paired samples from the same patient before (PRE) and after (POST) anthracycline chemotherapy treatment were included (n = 12). Results showed that VEGFR3 became significantly overexpressed after anthracycline-based chemotherapy (*p* = 0.0034) (Figure 3A). Through KMPlotter, the prognostic value of VEGFR3 was assessed in a cohort of 131 BC basal patients treated with chemotherapy: high expression of the gene was statistically correlated with low relapse-free survival (*p* = 0.00081), thus suggesting VEGFR3 as a bad prognostic factor (Figure 3B).

## 3. Discussion

Cytotoxic chemotherapy continues to be used for BC treatment, alone or in combination with other drugs. Particularly in TNBC, a subtype of BC that lacks a specific target, chemotherapy is the main treatment. Despite the side effects of this type of drug, the balance of benefit vs. risk is positive, supporting its continued use. However, efforts to minimize the negative effects of chemotherapy and to improve its efficacy are necessary. In this sense, there are indications that chemotherapy could modulate the microenvironment in a way favoring the propagation of the tumor. Some authors suggest a role of chemotherapy in the induction of cancer progression, besides the harmful effect on tumor cells. On one side, chemotherapy seems to enhance the mesenchymal and cancer stem cell features of the tumor cells. For example, in the MCF7 cell line, the long-term treatment with paclitaxel or doxorubicin induces expression of vimentin and N-cadherin, decreases E-cadherin, and increases the expression of C-myc and Oct4, as well as the secretion of VEGF factors [20]. On the other hand, chemotherapy increases the immunosuppressive capacity of cancer cells and improves their ability to organize into capillary-like structures [20]. The overexpression of the lymphangiogenic factors VEGFC and VEGFD, activators of VEGFR3, has been associated with lymph node metastasis and poor outcome in BC patients [21,22]. Moreover, the tumor-associated lymphatic vessels can upregulate the programmed death-ligand 1 (PDL1), thus reducing the anti-tumor response by avoiding the activation of T cells. These pieces of evidence suggest that lymphangiogenesis could play a key role in the induction of immune tolerance.

In this study, we have evaluated if doxorubicin, a common chemotherapy treatment, modifies the lymphangiogenic behavior of BC cells. We have focused on VEGFR3 in the TNBC subtype. Our experiments showed a significant increase in VEGFR3 expression at mRNA and protein levels after short-term treatment with doxorubicin in cellular models of TNBC. In contrast, the VEGFR3 increment did not happen in the luminal BC model. Those differences can be explained by intrinsic TNBC behavior. Indeed, different BC subtypes display differential lymphangiogenic signatures both in tumor cells and in the tumor microenvironment [8,9].

Published results have shown that positive expression of VEGFD detected by immunohistochemistry in BC is associated with an increased risk of relapse and a worse response to treatment [23]. Furthermore, VEGFR3 overexpression or activation has been proven to confer chemoresistance in others types of cancer such as leukemia by enhancing Bcl-2 expression [24]. In order to assess the possible involvement of VEGFR3 in the doxorubicin resistance mechanisms, we analyzed its expression in a doxorubicin-induced resistant cell line. This cell line was generated by subjecting the parental cell line to intermittent treatment with doxorubicin for 12 months. The resistant cells showed higher mRNA and protein levels of VEGFR3 when compared to the parental cell line. VEGFR3 overexpression has been shown to promote BC proliferation, migration, and cell survival in vitro, and increases tumor formation in vivo [13]. The downregulation of this receptor gives rise to metastasis reduction in in vivo models of cancer [25,26,27]. In concordance, our experiments showed that VEGFR3 silencing reduced the proliferation and migration capacities of BC cancer cells. This reduction was higher in parental cells than in doxorubicin-resistant cells that expressed elevated VEGFR3. The combination of VEGFR3 silencing with doxorubicin treatment increased the effect of the drug in parental cells, but not in resistant cells. However, VEGFR3 silencing in the resistant cell line was able to counteract the innate proliferation and migration capabilities of that cell line.

The suggested counter-therapeutic effect of doxorubicin has also been described for docetaxel in BC [15]. An upregulation of pro-lymphangiogenic factors, an increase in tumor invasion and metastasis, and, consequently, an increase in tumor survival and a reduction in drug efficacy have been described. Interestingly, this effect of docetaxel can be attenuated by blocking VEGFR3. Indeed, MAZ51, a small molecule inhibitor against VEGFR3, reduces cell invasion when combined with docetaxel [15]. It has also been described that chemotherapy treatment induces enrichment of the Notch signaling pathway as well as an increment in VEGF secretion. However, inhibition of the Notch pathway reduces VEGF secretion and the microvessel density in a BC xenograft tumor model. Specifically, it has been reported that Notch4 silencing could suppress VEGFR3 expression [20].

In addition to the higher VEGFR3 expression levels in the doxorubicin-resistant BC cell line, the analysis of BC samples showed significantly elevated VEGFR3 levels in relapsed compared to non-relapsed patients. Furthermore, high VEGFR3 expression was significantly positively correlated with worse survival in chemotherapy-treated basal BC patients. In fact, patients with unchanged or increased plasma VEGF and VEGFD levels after chemotherapy treatment show a worse response to treatment than those with reduced VEGF levels [23].

Even though the implication of VEGFR3 in doxorubicin resistance is questionable, it seems clear that elevated VEGFR3 levels reduce the efficacy of doxorubicin treatment. The data above suggest that the levels of this receptor could be a potential marker of poor response. Consequently, the combination of chemotherapy and VEGFR3 blockage could be a useful therapeutic strategy for the treatment of TNBC.

## 4. Materials and Methods

### 4.1. Cell Lines and Reagents

The MDA-MB-231, MDA-MB-468, and MCF7 BC cell lines were obtained from ATCC and cultured at 37 °C in a humidified 5% CO2, 95% air incubator. MDA-MB-231 and MDA-MB-468 are two triple-negative breast cancer cell lines from adenocarcinoma; MDA-MB-231 is characterized by *BRAF, CDKN2A, KRAS, NF2,* and *TP53* mutant genes, and MDA-MB-468 is characterized by *PTEN, RB1, SMAD4,* and *TP53* mutant genes. MCF7 is a luminal A breast cancer cell line from adenocarcinoma which is characterized by *PIK3CA, BCR, ADAM17,* and *VEGFC* mutant genes, among others. MDA-MB-231 cells and MDA-MB-468 cells were grown in Dulbecco’s Modified Eagle Medium F-12 (Invitrogen, Carlsbad, CA, USA) supplemented with 10% FBS and 1% penicillin–streptomycin. MCF7 cells were grown in Dulbecco’s Modified Eagle Medium F-12 (Invitrogen) supplemented with 10% FBS, 1% L-Glutamine, and 1% penicillin–streptomycin. Doxorubicin (Ferrer Farma, Barcelona, Spain) was used in all the cell cultures at 5 µM, during the indicated time periods, according to previous results [28,29,30,31]. The MDA-MB-231R (doxorubicin-resistant) cell line was generated by exposing the cells to increasing concentrations of doxorubicin, as previously reported, and grown in Dulbecco’s Modified Eagle Medium F-12 (Invitrogen) supplemented with 10% FBS and 1% penicillin–streptomycin [29].

### 4.2. MTT Proliferation Assay

Cell proliferation was measured using an MTT-based Cell Growth Determination Kit (# M5655; Sigma-Aldrich, St Louis, MO, USA). The MTT solution was added to each well in sterile conditions (final concentration was 10% of total volume), and the plates were incubated for 4 h at 37 °C. Purple formazan crystals were formed by succinate dehydrogenase in viable cells and were dissolved in a solubilization solution (1:1). The absorbance of the dissolved formazan product was measured at a 570 nm background corrected to 690 nm using a microplate reader.

### 4.3. Confocal Immunofluorescence Microscopy

For immunocytochemistry, cells were grown to 80% confluence in eight-well chamber slides (Corning Costar, Wiesbaden, Germany), fixed with 4% paraformaldehyde (Sigma-Aldrich), and subjected to immunostaining as follows: After permeabilization with Triton X-100 for 2 min and a 20 min blocking step with phosphate-buffered saline (PBS) containing 5% goat serum (Sigma-Aldrich), 1% BSA (Sigma-Aldrich), and 0.3% Triton X-100 (Sigma-Aldrich), the slides were washed and the following primary antibody was used in an overnight incubation at 4 °C at a dilution of 1:800 in 1% PBS, 0.3% BSA, and Triton X-100: mouse anti-VEGFR3 monoclonal antibody (#ab27278, Abcam, Cambridge, UK). The slides were washed three times for 5 min with PBS with 0.25% BSA and 0.1% Triton X-100 and incubated with Alexa–Fluor 647-conjugates goat anti-mouse IgG (Thermofisher, Waltham, MA, USA) secondary antibody in PBS with 0.25% BSA and 0.1% Triton X-100 at a dilution of 1:400 at room temperature for 1 h. Samples that were incubated without a primary antibody served as negative controls. Doxorubicin auto-fluorescent emission wavelength was described at ~590 nm. Slides were washed three times for 5 min with PBS with 0.25% BSA, 0.1% Triton X-100, and 0.05% Tween-20. Finally, the slides were incubated with 2 ug/mL of Hoechst for 10 min. Laser scanning microscopy was performed using a Leica TCS confocal microscope (Leica, Bensheim, Germany).

### 4.4. Real-Time Quantitative PCR Analysis

Specific Taqman probes for the target and the endogenous genes were purchased from Applied Biosystems (Waltham, MA, USA). Total RNA was isolated using the NucleoSpin RNA/Protein Kit (Macherey-Nagel, Düren, Germany) and reverse transcribed to cDNA using the cDNA Archive Kit (Applied Biosystems). cDNAs were combined with Taqman probes specific for each gene of interest along with a predeveloped Taqman Gene Expression Master Mix (Applied Biosystems). The RT-qPCR protocol was 50 °C for 2 min and 95 °C for 10 min followed by 40 cycles of 95 °C for 15 s and 60 °C for 1 min. Negative controls were included and yielded no products. RT-qPCR analysis was carried out on an ABI7900HT Fast Real-Time PCR System (Thermofisher). Ct values were determined using SDSv2.3 software (Applied Biosystems) and compared using the 2^−ΔΔCt^ method.

### 4.5. Transient Transfection of siRNA

The cell lines were transfected with VEGFR3 siRNA (100 nM) for 6 h in order to modulate mRNA expression levels, and the day after, they were exposed to 5 μM of doxorubicin for 24 h. siRNA inhibitors are single-stranded, modified RNAs that specifically inhibit endogenous mRNA molecules and cause a downregulation of mRNA activity. siRNA for VEGFR3 was purchased from Ambion (#AM16708, Austin, TX, USA). Scrambled siRNA (#AM17120, Ambion) was used as a negative transfection control. The reactions were performed with the Lipofectamin-2000 Transfection reagent (#11668019, Thermofisher), following the manufacturer’s instructions.

### 4.6. Wound-Healing Assay

An in vitro wound-healing/scratch assay was used to assess the capacity for tumor cell motility. Upon reaching confluency, cells were treated with siVEGFR3 and siSCR for 6 h, and the day after, MDA-MB-231 and MDA-MB-231R (3 × 10^5^ cells/well) were seeded in 24-well plates and cultured overnight. Then, they were sustained for 24 h either with doxorubicin 5 μM or complete media as a control. Finally, the cell monolayer was scratched with a sterile plastic tip and then immediately washed with PBS twice and cultured again in growing media in absence of serum, at 37 °C in a humidified incubator with 5% CO_2_. Wound-healing pictures per condition were obtained at 0 and 24 h after wound formation at 5X magnification

### 4.7. Western Blot and Immunoprecipitation

For protein extraction, cell monolayers were scraped into 1 mL of Pierce RIPA buffer (#89900, Thermofisher). The lysates were transferred to a clean microfuge tube, placed on ice for 15 min, sonicated for 30 s at 50% amplitude, and then centrifuged for 10 min at 14,000 rpm. The supernatant was transferred to a clean microfuge tube, and the protein concentration was determined. Protein extracts (40 μg) were boiled in Laemmli buffer and resolved on a 12% SDS–polyacrylamide gel, before being transferred onto a nitrocellulose membrane. Membranes were blocked in 5% BSA (Sigma-Aldrich) for 1 h and then incubated with antibodies for VEGFR2 (#ab39638, Abcam), VEGFR3 (#PA5-16871, Invitrogen), Tubulin (#sc-5286, Santa Cruz Biotechnology, Santa Cruz, CA, USA), and GAPDH (#MA5-15738, Invitrogen) overnight at 4 °C. In addition, proteins were precipitated with specific antibodies above mentioned and Protein G-plus agarose immunoprecipitation reagent (#sc-2002, Santa Cruz Biotechnology), washed three times with 1% Triton X-100 lysis buffer, and separated by sodium dodecyl sulfate–polyacrylamide gel electrophoresis (SDS-PAGE). The membranes were subsequently washed and then incubated for 1 h with an anti-mouse or anti-rabbit IgG horseradish peroxidase-linked secondary antibody (#7076 and #7074, Cell Signaling, Danvers, MA, USA). The membranes were then washed and briefly incubated using the Amersham ECL Western Blotting detection reagent (#RPN2209, GE Healthcare, Chicago, IL, USA).

### 4.8. Sample Patients

For neoadjuvant BC analysis, formalin-fixed and paraffin-embedded (FFPE) sample tissues were obtained from patients at Biomedical Research Institute INCLIVA (Spain) and were subsequently treated following standard guidelines. Twenty-four samples of twelve TNBC patients (Table 1) were selected to analyze the changes in expression of the VEGFR3 gene, between untreated tumor samples (biopsy) and anthracycline neoadjuvant treated samples (surgery).

Genetic material was isolated from FFPE tissue blocks using the RecoverAll Total Nucleic Acid Kit (Ambion). RNA was extracted from manually microdissected areas of 4 tissue sections (10 μm thick) on glass slides selected by a pathologist for each relevant FFPE tissue block. For standard mRNA analysis, 1 μg of total RNA concentration measured using a NanoDrop 2000 Spectrophotometer (Thermofisher) was reverse transcribed with random primers using the High-Capacity cDNA Reverse Transcription Kit (Applied Biosystems) and 5 ng of cDNA from each FFPE tissue and was then analyzed by RT-qPCR. In the case of pre-amplification, 25 ng of total RNA from FFPE tissue blocks was reverse transcribed, pre-amplified for 14 cycles using the 2X TaqMan PreAmp Master Mix (Applied Biosystems) according to manufacturer’s instructions, and diluted 1:5 before RT-qPCR analysis (described above). Ethical approval for the study was obtained from the Research Ethics Committee of the Hospital Clínico Universitario de Valencia (Spain) (Reference 2014/178, approved 25 June 2015). All patients signed written informed consent for study enrollment.

### 4.9. Survival Analysis

Kaplan–Meier plotter (KMplotter) tool (https://kmplot.com/analysis/) was used to evaluate the predictive/prognostic value of VEGFR3 on patient survival. By entering the mRNA IDs of interest into the field of the website, BC patients from GEO datasets (METABRIC study) were divided into two groups according to the expression level of the mRNA with auto-selected best cut-off, and the relapse-free survival was statistically analyzed. The hazard ratio (HR) with 95% confidence intervals and log-rank *p*-value were calculated and shown. The obtained results were used to identify the distinct prognostic values of VEGFR3 on TNBC after chemotherapy treatment.

### 4.10. Statistical Analyses

Each experiment was performed in technical and biological triplicate, and statistically significant differences were determined using GraphPad Prism 6.0. All data were presented as mean ± SD. Mean comparisons were performed using two-tailed Student’s *t*-test for normal distribution and Mann–Whitney U test for abnormal distribution. *p* < 0.05 was considered statistically significant.

## Figures and Tables

**Figure 1 ijms-24-03601-f001:**
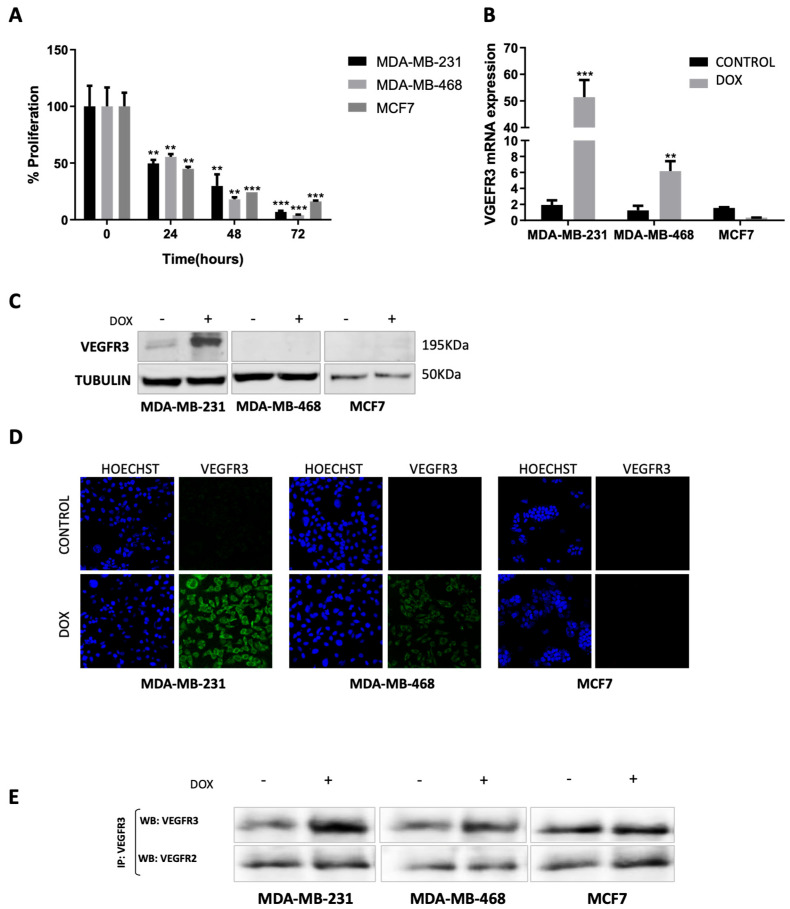
Effect of doxorubicin treatment in MDA-MB-231, MDA-MB-468, and MCF7 breast cancer cell lines. Viability of MDA-MB-231, MDA-MB-468, and MCF7 cell lines after treatment with 5 μM doxorubicin for 24, 48, and 72 h was measured through MTT assay. The analysis was performed by comparing each time point with the initial one (t0) (**A**). mRNA levels of VEGFR3 at control condition and after treatment with doxorubicin (5 µM for 24 h) measured by RT-qPCR in MDA-MB-231, MDA-MB-468, and MCF7 cells. GAPDH was used as endogenous gene. The statistical analysis was performed by comparing untreated vs. treated condition in each cell line (**B**). VEGFR3 protein expression with or without 24 h 5 uM doxorubicin treatment was assessed by Western blot in MDA-MB-231, MDA-MB-468, and MCF7 cells. α-tubulin was used as endogenous control (**C**). Representative immunofluorescence images for VEGFR3 (green) and Hoescht (blue) in MDA-MB-231, MDA-MB-468, and MCF7 cells after doxorubicin (5 µM for 24 h) treatment and control (no treatment) (**D**). Immunoprecipitation with anti-VEGFR3 antibody and Western blot with anti-VEGFR3 and anti-VEGFR2 antibodies in MDA-MB-231, MDA-MB-468, and MCF7 cells (**E**). ** *p* < 0.01, *** *p* < 0.001. DOX: doxorubicin.

**Figure 2 ijms-24-03601-f002:**
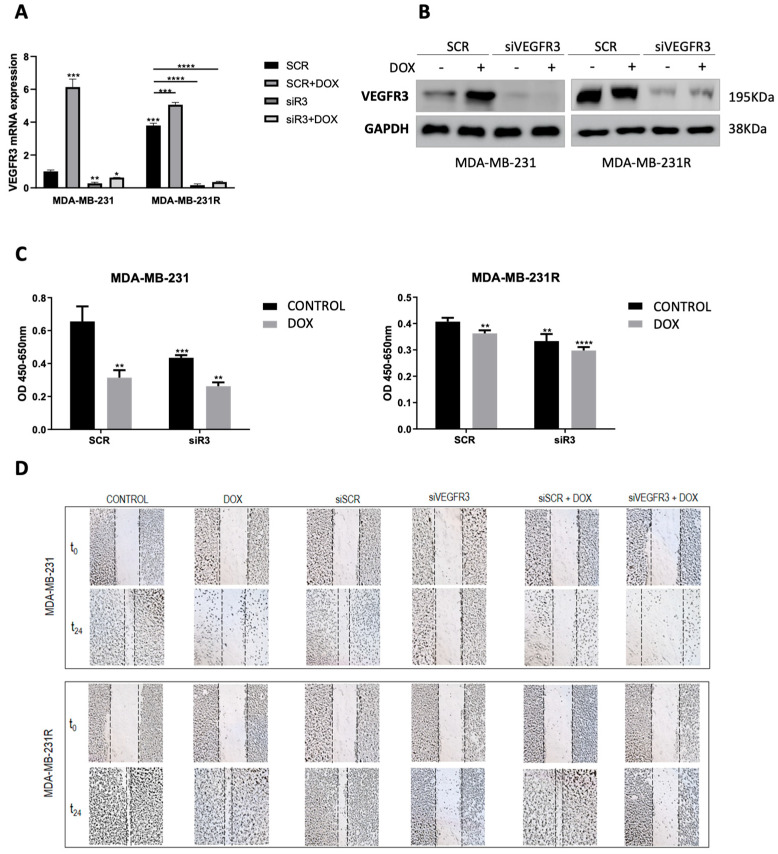
Effect of doxorubicin treatment in MDA-MB-231 cells vs. MDA-MB-231R cells. VEGFR3 mRNA expression after MDA-MB-231 or MDA-MB-231R transfection with silencer control (MDA-MB-231 SCR and MDA-MB-231R SCR) or VEGFR3 silencer (MDA-MB-231 siR3 and MDA-MB-231R siR3) with or without doxorubicin treatment (DOX). Gene expression was evaluated by RT-qPCR, and GAPDH was used as endogenous gene. The statistical analysis was performed by comparing each condition to SCR control in each cell line (**A**). VEGFR3 protein expression was evaluated by Western blot in the same conditions for MDA-MB-231 and MDA-MB-231R cell lines. GAPDH was taken as endogenous control (**B**). Viability of MDA-MB-231 and MDA-MB-231R cells was evaluated by MTT assay when VEGFR3 was silenced (siVEGFR3) and after doxorubicin (DOX) treatment. (**C**). Wound-healing assay was performed to evaluate the migration capacity of MDA-MB-231 and MDA-MB-231R cells in the same conditions described above (5× magnification) (**D**). * *p* < 0.05, ** *p* < 0.01, *** *p* < 0.001, **** *p* < 0.0001. DOX: doxorubicin.

**Figure 3 ijms-24-03601-f003:**
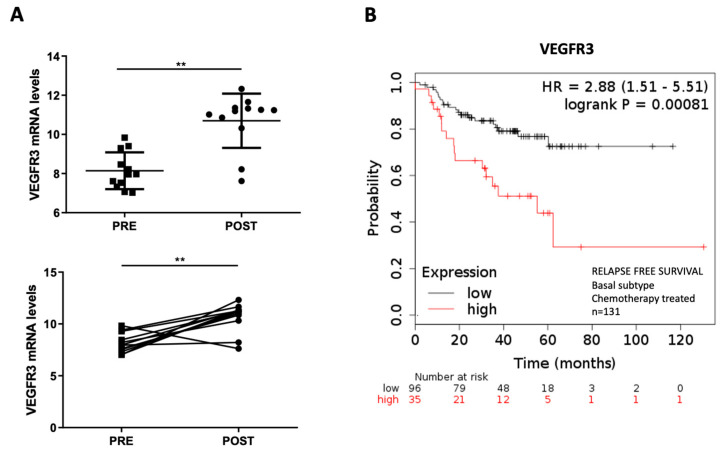
VEGFR3 expression analysis and prognostic value in triple-negative breast cancer patients. The VEGFR3 mRNA expression was analyzed by RT-qPCR in 24 samples from anthracycline-treated TNBC patients. Paired samples from the same patient before (PRE, n = 12) and after (POST, n = 12) anthracycline chemotherapy treatment were included. GAPDH was used as endogenous control. ** *p* < 0.01 (**A**). A cohort of 131 BC basal patients treated with chemotherapy was analyzed using the Kaplan–Meier Plot tool with a 120-month follow-up for VEGFR3 mRNA with auto-selected best cut-off for low (black) and high (red) expression (**B**).

**Table 1 ijms-24-03601-t001:** Summary of clinical–pathological patient characteristics.

Sample	Subtype	Diag. Age	Diagnosis	Grade	RCB	Chemotherapy
n1	TN	41	ICD	II	1	Taxolx12+FEC-100x4
n2	TN	70	ICD	III	1	Taxolx12+FEC-100x4
n3	TN	78	ICD	III	1	Atx4+CMFx4
n4	TN	68	ICD	II	1	Atx4+CMFx4
n5	TN	42	ICD	II	2	Atx4+CMFx4
n6	TN	64	ICD	II	2	Taxolx12+FEC-100x4
n7	TN	43	ICD	III	2	Taxolx12+FEC-100x4
n8	TN	45	ICD	II	2	Atx4+CMFx4
n9	TN	52	ICD	II	3	Taxolx12+FEC-100x4
n10	TN	46	ICD	III	3	Taxolx12+FEC-100x4
n11	TN	65	ILC	II	3	Atx4+CMFx4
n12	TN	52	ICD	II	3	Taxolx12+FEC-100x4

TN: triple-negative, Diag age: age at diagnosis, ICD: infiltrating ductal carcinoma, ILC: infiltrating lobular carcinoma, RCB: residual cancer burden, FEC-100: (5-fluorouracil/epirubicin/cyclophosphamide), CMF: (cyclophosphamide/methotrexate/5-fluorouracil), At: (doxorubicin/paclitaxel).

## Data Availability

Not applicable.

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
