# Peer review of "High VEGFR3 Expression Reduces Doxorubicin Efficacy in Triple-Negative Breast Cancer"

_ijms, 2023, doi:10.3390/ijms24043601_

Round 1
Reviewer 1 Report
Dear Authors;
I reviewed your paper and I found it very interesting with the scientific results to be published. But, there is a minor correction in the following text:
Line 89: Explanation of “D” in "Figure. 1", is substituted with “E”. And explanation for “D” should be added. Best RegardsAuthor Response
Thanks for the reviewers' comments and suggestions. We have answered one by one the questions raised.
Reviewer 1
Dear Authors;
I reviewed your paper and I found it very interesting with the scientific results to be published. But, there is a minor correction in the following text: Line 89: Explanation of “D” in "Figure. 1", is substituted with “E”. And explanation for “D” should be added. Best Regards
We have corrected this incident. Inserted changes are highlighted in yellow in manuscript.
Reviewer 2 Report
The reviewed manuscript is an interesting item for scientists dealing with the biology of breast cancer. However, I do have some significant remarks. They mainly concern the method of presenting research results.
1. Poor image quality in Western Blot assays. Was the study performed in 3 independent biological replicates? Deviations for the MDA-MB-231 line seem extremely small for the image quality.
2. It would be valuable if the presentation of the results of the scratch healing test were supplemented with quantitative results.
e.g. https://www.ncbi.nlm.nih.gov/pmc/articles/PMC5691760/
3. There is an error in the description in Figure 1 - there is no description of the results of the study using a confocal microscope
4. molecular characterization of selected cell lines is missing, especially since the MDA-MB-231 and MDA-MB-436 lines, although both are TNBC lines, have completely different molecular profiles
Author Response
The reviewed manuscript is an interesting item for scientists dealing with the biology of breast cancer. However, I do have some significant remarks. They mainly concern the method of presenting research results.
- Poor image quality in Western Blot assays. Was the study performed in 3 independent biological replicates? Deviations for the MDA-MB-231 line seem extremely small for the image quality.
We understand the doubt raised. All western blots have been made in triplicate, and although it is true that the image is of improvable quality, the deviation obtained after quantifying the three is very small in this case.
- It would be valuable if the presentation of the results of the scratch healing test were supplemented with quantitative results. e.g. https://www.ncbi.nlm.nih.gov/pmc/articles/PMC5691760/
We have taken into account your suggestion and have carried out the quantification corresponding to the wound-healing test. You can see this as supplementary figure 1, and the changes corresponding to the introduction of the p values are highlighted in yellow in the text.
- There is an error in the description in Figure 1 - there is no description of the results of the study using a confocal microscope
It is true, we have corrected this incident. Inserted changes are highlighted in yellow
- molecular characterization of selected cell lines is missing, especially since the MDA-MB-231 and MDA-MB-436 lines, although both are TNBC lines, have completely different molecular profiles
We have taken into account this suggestion and a brief molecular characterization description of the breast cancer cell lines used are specified in materials and methods (cell lines and reagents). Insert changes are highlighted in yellow.

Reviewer 3 Report
Major Concerns
- TNBC is widely considered a heterogeneous disease and is subdivided into molecularly characterized subtypes. Basal TNBC differs from Mesenchymal TNBC subtypes. The chosen cell lines fall into these different TNBC subtypes and are differential in their sensitivity to doxorubicin and their VEGFR expression, as indicated in the paper. More examples of these subtypes would allow possible correlations with specific TNBC subtypes.
VEGF/VEGFR have long been studied in the context of breast cancer, TNBC, and TNBC subtypes.
- VEGFR3, largely expressed in lymphatic endothelial cells, has been shown to be overexpressed in tumors, is only "associated" with tumor cell proliferation, migration and invasion. Therefore, studying it in tumor cell lines in culture is difficult to correlate with patient or animal tumor models that contain tumor vasculature and the tumor microenvironment in addition to the tumor cells and would likely provide different outcomes to treatments. At the minimum, it should be shown that a combination of a VEGFR3 inhibitor would additively or synergistically be used with doxorubicin, especially in the doxo-resistant MDA-MB-231 cell line. In addition, does overexpression of VEGFR3 in MDA-MB-468 lead to resistance to doxorubicin?
TNBC cell line VEGF expression and doxorubicin treatment data is publicly available and should be used to correlate expression of VEGFR3 with Doxo response. Combination of VEGFR inhibitor + doxo in a select few lines would further strengthen the findings and perhaps suggest a TNBC subtype specific treatment strategy. Showing xenografts of single and combination treatments would further show in vivo efficacy.
Minor concerns
- MCF7 cell lysate is significantly underloaded (tubulin) compared to the TNBC lines in Western blot in Fig 1C and therefore hard to conclude if VEGFR3 is expressed or induced with doxo.
- If VEGFR3 is overexpressed in MDA-MB-231 with doxo treatment but not significantly increased heterodimerization with VEGFR2, is the excess VEGFR3 not functional in the tumor cells?
- VEGFR3 expression by Western Blot (Fig 1C) doesn't correlate very well with expression by IF (Fig 1D), in particular in MDA-MB-468. Are these different antibodies? If the same, why the discrepancy?
- How are "basal" BCs defined in the clinical cohort? Not all basal BC are TNBCs and not all TNBCs are basal (~30-50%).
- Clinical TNBC samples were from patients treated with combination chemo, of which only a few included doxorubicin. Is the expression of VEGFR3 (in 231 cells or resistant 231 cells also cross-resistant with taxanes? with CMF or FEC treatments? Can this "resistance" be correlated by VEGFR3 expression alone or is it simply one of several markers of a mesechemal/stem-like subtype that is inherently resistant to chemo agents such as doxo?
Author Response
Major Concerns
- TNBC is widely considered a heterogeneous disease and is subdivided into molecularly characterized subtypes. Basal TNBC differs from Mesenchymal TNBC subtypes. The chosen cell lines fall into these different TNBC subtypes and are differential in their sensitivity to doxorubicin and their VEGFR expression, as indicated in the paper. More examples of these subtypes would allow possible correlations with specific TNBC subtypes.
The reviewer is right, not all triple negative breast cancer cell lines show exactly the same behavior. In the present manuscript we chose two cell lines with different characteristics as has been included in the material and methods of the current version (highlighter in yellow).
VEGF/VEGFR have long been studied in the context of breast cancer, TNBC, and TNBC subtypes.
The VEGF/VEGFR pathway has been extensively studied in the breast cancer context. However, less is known about lymphangiogenic factors and receptors. Specifically, there is lack of information on the possible involvement of doxorubicin in the lymphangiogenic mechanism, a subject where our work has been mainly focused.
- VEGFR3, largely expressed in lymphatic endothelial cells, has been shown to be overexpressed in tumors, is only "associated" with tumor cell proliferation, migration and invasion. Therefore, studying it in tumor cell lines in culture is difficult to correlate with patient or animal tumor models that contain tumor vasculature and the tumor microenvironment in addition to the tumor cells and would likely provide different outcomes to treatments. At the minimum, it should be shown that a combination of a VEGFR3 inhibitor would additively or synergistically be used with doxorubicin, especially in the doxo-resistant MDA-MB-231 cell line. In addition, does overexpression of VEGFR3 in MDA-MB-468 lead to resistance to doxorubicin?
We appreciate the commentaries of the reviewer. Sure, the in vitro experiments lose microenvironmental and vasculatures elements relevant for a global cancer view. However, we considered a cell line a good first approach as a prove of concept. We show that VEGR3 is implicated in cellular proliferation and migration, and its levels of expression could be an indicator of different doxorubicin response.
We reinforce our result with data base information and bibliographic data in others model including xenograft breast cancer. We will consider the in vivo approach for future validation of those results.
TNBC cell line VEGF expression and doxorubicin treatment data is publicly available and should be used to correlate expression of VEGFR3 with Doxo response. Combination of VEGFR inhibitor + doxo in a select few lines would further strengthen the findings and perhaps suggest a TNBC subtype specific treatment strategy. Showing xenografts of single and combination treatments would further show in vivo efficacy.
The Kaplan-Meier figure (Fig. 3B) shows the correlation between the expression of VEGFR3 in TNBC patients treated with chemotherapy and the follow-up of the patients. This information has been obtained from public databases. Regarding the cell lines, we have selected two different TNBC cell lines for our work. Increasing the number of TNBC cell lines, as well as in vivo assays, will strengthen our results and we consider them for future validation.
Minor concerns
- MCF7 cell lysate is significantly underloaded (tubulin) compared to the TNBC lines in Western blot in Fig 1C and therefore hard to conclude if VEGFR3 is expressed or induced with doxo.
It is true that the endogenous control tubulin does not present the same intensity as the other cell line. However, this western was performed in triplicate and we show the quantification of these replicates, which are all referenced to their respective endogenous control. We achieve this by dividing the intensity of the corresponding band (VEGFR3 in this case) with respect to the intensity of tubulin.
- If VEGFR3 is overexpressed in MDA-MB-231 with doxo treatment but not significantly increased heterodimerization with VEGFR2, is the excess VEGFR3 not functional in the tumor cells?
The excess of VEGFR3 may be functional as VEGFR3 forms homodimers in addition to heterodimers with VEGFR2 [1].
- VEGFR3 expression by Western Blot (Fig 1C) doesn't correlate very well with expression by IF (Fig 1D), in particular in MDA-MB-468. Are these different antibodies? If the same, why the discrepancy?
The antibodies against VEGFR3 for immunofluorescence and western-blot are different (see references for materials and methods). This fact, together with the different sensitivity of the techniques used, may explain the divergences of expression in MDA-MB-468.
- How are "basal" BCs defined in the clinical cohort? Not all basal BC are TNBCs and not all TNBCs are basal (~30-50%).
Classification was based on the St. Gallen consensus. Immunohistochemistry-based St. Gallen surrogates have been confirmed to adequately represent molecular subtypes. However, the classification based on arrays expression matrices differs from the classification based on the expression of estrogen receptors, progesterone receptors, HER2 and Ki67 [2]
- Clinical TNBC samples were from patients treated with combination chemo, of which only a few included doxorubicin. Is the expression of VEGFR3 (in 231 cells or resistant 231 cells also cross-resistant with taxanes? with CMF or FEC treatments? Can this "resistance" be correlated by VEGFR3 expression alone or is it simply one of several markers of a mesechemal/stem-like subtype that is inherently resistant to chemo agents such as doxo?
The treatment of cell lines with combined drugs such as CEF and FEC, and the generation of resistance models will be more difficult to perform, but anyway interesting to prove the particular relationship with VEGFR3.
The reason for using these patients’ samples was the availability of pre- and post- treatment samples that could give us an idea about receptor regulation after treatment.
Moreover, data showing that VEGFR3 overexpression or activation confer chemoresistance in others types of cancer such as leukemia reinforce our hypothesis in BC [24]. In addition, docetaxel upregulates pro-lymphangiogenic factors, increases tumor invasion and metastasis and, consequently, increases tumor survival and reduces drug efficacy in BC. This effect of docetaxel can be attenuated by blocking VEGFR3. These data suggest that VEGFR3 as a general mechanism after chemotherapy treatment, however it needs further confirmation with future projects [3].
References
- Monaghan, R.M.; Page, D.J.; Ostergaard, P.; Keavney, B.D. The Physiological and Pathological Functions of VEGFR3 in Cardiac and Lymphatic Development and Related Diseases. Cardiovasc Res 2021, 117, 1877–1890, doi:10.1093/cvr/cvaa291.
- Vasconcelos, I.; Hussainzada, A.; Berger, S.; Fietze, E.; Linke, J.; Siedentopf, F.; Schoenegg, W. The St. Gallen Surrogate Classification for Breast Cancer Subtypes Successfully Predicts Tumor Presenting Features, Nodal Involvement, Recurrence Patterns and Disease Free Survival. The Breast 2016, 29, 181–185, doi:10.1016/j.breast.2016.07.016.
- Harris, A.R.; Perez, M.J.; Munson, J.M. Docetaxel Facilitates Lymphatic-Tumor Crosstalk to Promote Lymphangiogenesis and Cancer Progression. BMC Cancer 2018, 18, 718, doi:10.1186/s12885-018-4619-8.
Reviewer 4 Report
In the present manuscript Torres-Ruiz S. et al.evaluate the expression of VEGF3 in basal TNBC and after doxorrubicin treatment. Authors proposed that the levels of this receptor could be a potential marker of doxorubicin response. The article is well organized and have merit. The following issues need attention and pherphaps adittional experiments to validate the main idea of the work.
1. Relevant references related alternative TNBC treatment options could be incorporated in the section.doi: 10.3390/biomedicines10051130, doi: 10.3390/pharmaceutics14030626 , doi: 10.3390/ijms23031665
2. In the result section, page 2 line 86, the following statement “These results suggest that doxorubicin may have a role in the lymphangiogenic process” needs experimental support. In vitro evaluation of lymphangiogenesis could be include to validate hypothesis. Please read the following article https://doi.org/10.3389/fbioe.2021.697657
3. In figure 1c, tubulin control of MCF7 is not similar in intensity than the others cell lines. Bases in this, it is hard to make a comparation among cell lines.
4. Description of figure 1e is missing. Besides, proteín loading control is missing. Please check.
5. In the result section, page 5 line 140, authors mentione that resistant cells showed a 40% decrease after dox treatment before VEGFR3 silencing. OD showed in figure 2b is mismatched.
6. Wound healing figure in MDA cells after siVEGFR3 + dox or SCRB + dox are not well interpretated. How do the authors know that the effect is in the migration and not in proliferation?
7. A molecular mechanims would be interesting to investigate in relation of cell signaling after dox treatment and VEGFR3.
Author Response
In the present manuscript Torres-Ruiz S. et al. evaluate the expression of VEGF3 in basal TNBC and after doxorubicin treatment. Authors proposed that the levels of this receptor could be a potential marker of doxorubicin response. The article is well organized and have merit. The following issues need attention and perhaps additional experiments to validate the main idea of the work.
- Relevant references related alternative TNBC treatment options could be incorporated in the section.doi: 10.3390/biomedicines10051130, doi: 10.3390/pharmaceutics14030626, doi: 10.3390/ijms23031665
Following the reviewer's suggestion, we have added references related to alternative therapeutic treatments in TNBC. Highlighted in yellow.
- In the result section, page 2 line 86, the following statement “These results suggest that doxorubicin may have a role in the lymphangiogenic process” needs experimental support. In vitro evaluation of lymphangiogenesis could be include to validate hypothesis. Please read the following article https://doi.org/10.3389/fbioe.2021.697657
In order to clarify the conclusions, we have modified this sentence according to the results obtained. This modification is highlighted in yellow. The lymphangiogenic test would of course reinforce the results presented, but due to the limited time available we have not been able to perform it.
- In figure 1c, tubulin control of MCF7 is not similar in intensity than the others cell lines. Bases in this, it is hard to make a comparation among cell lines.
We have understood the doubt formulated and it is true that the endogenous control tubulin does not present the same intensity as the other cell lines. However, this western was performed in triplicate and we show the quantification of these replicates, which are all referenced to their respective endogenous control. We achieve this by dividing the intensity of the corresponding band (VEGFR3 in this case) with respect to the intensity of tubulin. On the other hand, the VEGFR3 band for the MCF7 line is negative, thus facilitating the comparison between lines.
- Description of figure 1e is missing. Besides, protein loading control is missing. Please check.
We have corrected this incident. The inserted changes are highlighted in yellow. Regarding figure 1e, it corresponds to a co-immunoprecipitation assay in which we precipitated the VEGFR3 protein alone with a specific antibody, thus missing GAPDH, tubulin and other loading controls. In that case, VEGFR2 is also measured, as it is known to complex with VEGFR3, with no changes detected.
- In the result section, page 5 line 140, authors mentioned that resistant cells showed a 40% decrease after dox treatment before VEGFR3 silencing. OD showed in figure 2b is mismatched.
We have corrected this incident. MDA-MB-231 cells decreased around 52% cell proliferation after doxorubicin treatment, while only a 11% of reduction is observed in resistant cells. Inserted changes are highlighted in yellow.
- Wound healing figure in MDA cells after siVEGFR3 + dox or SCRB + dox are not well interpretated. How do the authors know that the effect is in the migration and not in proliferation?
We understand the doubt raised. In that case, wound-healing assays were performed in absence of serum in order to avoid cell proliferation and promote only cell migration. This is in accordance with different articles in which this methodology is detailed [1,2]. This has also been indicated in materials and methods (Wound-healing assay) and is highlighted in yellow.
- A molecular mechanism would be interesting to investigate in relation of cell signaling after dox treatment and VEGFR3.
Undoubtedly, it would be very interesting to know the molecular mechanism after treatment with doxorubicin and VEGFR3. In this sense, it has been described that the treatment of breast cancer cells with chemotherapy (paclitaxel or doxorubicin) induces an enrichment of the Notch signaling pathway detected by RNA sequencing. Furthermore, inhibition of the Notch pathway results in reduced VEGF secretion and significantly lower microvessel density in a breast cancer xenograft tumor model. Specifically, it has been reported that Notch4 silencing could suppress VEGFR3 expression [3]. We have added an explanatory sentence in the discussion.
References
- Liang, C.-C.; Park, A.Y.; Guan, J.-L. In Vitro Scratch Assay: A Convenient and Inexpensive Method for Analysis of Cell Migration in Vitro. Nat Protoc 2007, 2, 329–333, doi:10.1038/nprot.2007.30.
- Rodriguez, L.G.; Wu, X.; Guan, J.-L. Wound-Healing Assay. Methods Mol Biol 2005, 294, 23–29, doi:10.1385/1-59259-860-9:023.
- Zhang, P.; He, D.; Chen, Z.; Pan, Q.; Du, F.; Zang, X.; Wang, Y.; Tang, C.; Li, H.; Lu, H.; et al. Chemotherapy Enhances Tumor Vascularization via Notch Signaling-Mediated Formation of Tumor-Derived Endothelium in Breast Cancer. Biochem Pharmacol 2016, 118, 18–30, doi:10.1016/j.bcp.2016.08.008.
Round 2
Reviewer 2 Report
The authors responded to all my comments and made some corrections/supplements to the results. I am satisfied with it and I believe that the work deserves to be published.Author Response
Thanks for your comments and suggestions.
Reviewer 3 Report
The concerns were addressed in an appropriate manner from the previous review and are sufficient for publication.
Author Response
Thanks for your comments and suggestions
Reviewer 4 Report
The authors have reviewed their article meticulously and have taken my suggestions into account. I have no more important suggestions.
Author Response
Thanks for your comments and sugggestions